# Quantifying the Effect of Supplementation with Algae and Its Extracts on Glycolipid Metabolism: A Meta-Analysis of Randomized Controlled Trials

**DOI:** 10.3390/nu12061712

**Published:** 2020-06-08

**Authors:** Kun-xiang Ding, Tian-lin Gao, Rui Xu, Jing Cai, Hua-qi Zhang, Yong-ye Sun, Feng Zhong, Ai-guo Ma

**Affiliations:** 1School of Public Health, Qingdao University, Qingdao 266021, China; dkunxiang@163.com (K.-x.D.); gaotl@qdu.edu.cn (T.-l.G.); xurui9510@163.com (R.X.); 15165231565@163.com (J.C.); huaqi_erin@163.com (H.-q.Z.); yongye.sun@126.com (Y.-y.S.); magfood@126.com (A.-g.M.); 2Institute of nutrition and health, Qingdao University, Qingdao 266021, China

**Keywords:** algae, glycolipid metabolism, blood glucose, lipid profiles, meta-analyses

## Abstract

Aims: The effect of algae and its extract supplementation on glycolipid metabolism has not been finalized. Therefore, the purpose of the meta-analyses was to assess the effects of its supplementation on glycolipid metabolism concentration. Methods: We have systematically searched PubMed, Web of Science, the Cochrane Library and Embase to identify randomized controlled trials (RCTs) that investigated the impact of algae and its extracts supplementation on glycolipid metabolism. Effect size analysis was performed using weighted mean difference (WMD) and 95% CI between the methods of the experiment group and the control group. Subgroup analyses were performed to explore the possible influences of study characteristics. Publication bias and sensitivity analysis were also performed. Results: A total of 27 RCTs (31 trials) with 1221 participants were finally selected for the meta-analysis. The algae and its extract intervention significantly decreased glycosylated hemoglobin (HbA1c, WMD = −0.18%; 95% CI: −0.27 to −0.10; *p* < 0.001), high-density lipoprotein cholesterol (HDL-C, WMD = −0.22 mmol/L; 95% CI: −0.38 to −0.06; *p* = 0.008), and triglycerides (TC, WMD = −0.31 mmol/L; 95% CI: −0.37 to −0.25; *p* < 0.001) levels and increased insulin (WMD = 6.05 pmol/mL; 95% CI: 4.01 to 8.09; *p* < 0.001) levels. It did not significantly change the blood glucose, homeostasis model assessment-insulin resistance index (HOMA-IR), 2-h post-meal blood glucose (2hPBG) and other lipid profiles. Subgroup analyses based on the duration of intervention and subjects demonstrated that the intervention of algae and its extracts for 10 weeks or fewer and more than 40 subjects decreased TC levels (*p* < 0.05). Moreover, the intervention reduced TC and 2hPBG concentrations for East Asians (*p* < 0.05). Conclusions: Our findings provided evidence that algae and its extract interventions were beneficial for the regulation of human glycolipid metabolism. More precise RCTs on subjects are recommended to further clarify the effect of algae, seaweed polysaccharide, seaweed polypeptide, algae polyphenol and its products intervention on glycolipid metabolism.

## 1. Introduction 

In recent years, with the rapid development of the global economy and the increase of unhealthy lifestyles, diabetes and cardiovascular diseases have become the most common diseases with the highest mortality rates in the world [1,2]. According to the World Health Organization, more than 17.3 million people die of cardiovascular disease each year, and it is expected that the number of cardiovascular disease-related deaths will increase to more than 23.6 million by 2030 [3]. Atherosclerosis (AS) is the main pathological basis of cardiovascular diseases [4,5]. During the occurrence and development of AS, abnormal metabolism of neutral lipids, especially cholesterol and apolipoproteins, is the main factor leading to the occurrence and development of cardiovascular diseases [6,7,8]. Diabetes mellitus (DM) is a chronic disease, a condition caused either by insufficient insulin secretion or insulin resistance [9] characterized by hyperglycemia, and is widely prevalent worldwide and usually accompanied by impaired glucose tolerance (IGT), hypertension and hyperlipidemia [10]. In 2017, approximately 451 million adults worldwide had diabetes, and this number is expected to increase to 693 million by 2045 [11]. Abnormal glucose metabolism and lipid metabolism often occur in parallel. Therefore, blood glucose control and lipid control treatment should be carried out simultaneously.

Seaweed plays an important role in regulating chronic diseases because of its unique biologically active compounds, such as fucoidan, alginate, fucosterol, phlorotannins and phycocyanin, which are not found in terrestrial plant sources [12,13,14]. It can regulate intestinal health and reduce risk factors of diabetes, antiviral, anticancer, anticoagulation, etc. [15]. Dietary studies in Japan and South Korea showed that consumption of seaweed can reduce the incidence of chronic diseases such as cancer, hyperlipidemia and coronary heart disease [16,17]. The existing evidence shows that algae, especially spirulina, has antioxidant, anti-inflammatory, antitumor, antiviral, antibacterial and other health-promoting functions, and has a positive therapeutic effect on hyperlipidemia, obesity, cardiac vascular disease (CVD) and diabetes [15,18,19,20,21]. Polyphenols extracted from seaweed are thought to contribute to reducing the risk of cardiovascular disease and diabetes complications due to hyperglycemia, hyperlipidemia, oxidative damage and chronic inflammation, as well as metabolic abnormalities [22]. Recently, there have been many reports on the improvement of diabetes, obesity and hyperlipidemia by the bioactive ingredients in natural foods, especially seaweed [16,17,23,24,25]. However, other studies have shown that eating more seaweed increased the risk of metabolic syndrome [26]. Meanwhile, no meta-analysis has specifically pooled or summarized the precise effect of algae and its extract consumption on glycolipid metabolism. Therefore, we conducted a meta-analysis to investigate the impact of algae and its extract supplementation on concentrations for glycolipid metabolism in humans.

## 2. Methods

### 2.1. Search Strategy

We have systematically searched several databases, including PubMed, Web of Science, the Cochrane Library and Embase from inception to 1 December 2019. The search terms were as follows: (algae OR seaweed OR kelp OR laminaria japonica OR nori OR wakame OR undaria OR sea mustard OR sea lettuce OR sea kale OR nostoc OR gelidium OR hijiki OR sargassum fusiforme OR hizikia fusiforme OR gracilaria OR ulva clathrate OR spirulina OR chlorella OR algal polysaccharide OR trehalose OR fucoidan OR algae polyphenol OR algae polypeptide OR seaweed peptides) AND (FPG OR insulin OR HOMA-IR OR HbA1c OR HDL-C OR LDL-C OR Triglyceride OR Total cholesterol). All authors were involved in the screening of the inclusion trials.

### 2.2. Selection Criteria

The experiments that meet the requirements of the inclusion criteria were as follows: (1) Randomized controlled trials; (2) RCTs using algae, seaweed polysaccharide, algae polyphenol, algae polypeptide and its products as the intervention, and studies which were combined with other interventions were included when the control group received the same treatment; (3) RCTs using human clinical trials; (4) RCTs that provided information on baseline and post-intervention results for the experimental and control groups. The exclusion criteria were: (1) studies not related to the target research content; (2) studies lacking sufficient results; (3) reviews, letters, comments and abstracts.

### 2.3. Data Extraction

We extracted the following information from eligible articles: (1) first author’s name; (2) publication year; (3) study location; (4) number of participations in experimental and control groups; (5) trial design; (6) intervention duration; (7) daily dose and type of algae and its extracts; (8) age and gender of participants; (9) health status of subjects; (10) levels of lipid profile, FPG, 2hPBG, HOMA-IR, HbA1c and insulin.

### 2.4. Quality Assessment and Publication Bias

We used the Cochrane Collaboration’s tool for assessing risk of bias to assess the quality of selected RCTs. Potential publication bias was evaluated by Begg’s funnel plot asymmetry, Begg’s rank correlation and Egger’s weighted regression tests. Additional trim and fill analysis was then performed to test and adjust for publication bias [27].

### 2.5. Statistical Methods

Before the analysis, the study heterogeneity was tested using Cochrane’s Q test. An I^2^ ≥ 50% and/or a Q-statistic of *p* < 0.10 were evidence supporting the presence of heterogeneity [28], in which the random effects modeling method was needed. Otherwise, the fixed effects modeling method was applied. In RCTs reported levels of lipid profile, FBG, and 2hPBG in mg/dL, the data was converted to mmol/L before analyses, and for levels of insulin in μIU/mL, the data was converted to pmol/mL. Effect size of each study was calculated from mean and standard deviation (SD) of the results before and after the intervention and presented as weighted mean difference (WMD) and 95% confidence interval (CI). In studies that reported the standard error of the mean (SEM), SD was calculated as follows: SD = SEM × sqrt (*n*), where n is the number of subjects. Subgroup analyses were carried out based on intervention duration, sample size, intervention species, health status and area. The intervention duration and the sample size were divided into two subgroups with the boundary of 10 weeks and 40 participants, respectively; the types of interventions were divided into three groups according to spirulina, chlorella and other algae; health status was divided into health people, obesity, type 2 diabetes and other diseases; and the area was divided into three groups: East Asia, Southwest Asia, and non-Asia. Subgroup analysis is not performed on indicators with fewer than 3 studies. The results from our included studies were combined using Stata software version 11.0. *p* values are two tailed, and *p* < 0.05 was considered statistically significant.

## 3. Results

### 3.1. Search Results and Characteristics of Studies

A flow diagram showing the procedure of study selection is presented in Figure 1. From the electronic searches, 1409 potential literature citations and four additional records identified through other sources were identified. In the end, a total of 27 RCTs (30 trials) with 1221 participants were finally considered to be selected for the current meta-analysis [29,30,31,32,33,34,35,36,37,38,39,40,41,42,43,44,45,46,47,48,49,50,51,52,53,54,55]. The characteristics of RCTs included in the meta-analysis are summarized in Table 1. These included studies published between 1996 and 2019 and conducted in seven countries, including Japan, India, Korea, Mexico, America, Poland and Iran. The number of participants ranged from 12 to 80, and the intervention duration ranged from two weeks to 28 weeks. The subjects were pre-diabetic and patients with type 2 diabetes (T2D), obese or overweight subjects, healthy adults and patients with hyperlipidemia or hypercholesterolemia. Among the included studies, 22 trials used a placebo as the control; other studies used methods of controlling variables.

### 3.2. Effect of Algae and Its Extracts Intervention on Lipid Profiles

The meta-analysis was performed on data extracted from 23 RCTs for HDL-C (961 subjects), TG (869 subjects), and 25 RCTs for TC (979 subjects), LDL-C (980 subjects) (Figure 2). The results showed that algae intervention did not significantly change the LDL-C (WMD = 0.04 mmol/L; 95% CI: −0.16 to 0.24; *p* = 0.70; *I^2^* = 97%) and TG (WMD = −0.02 mmol/L; 95% CI: −0.36 to 0.32; *p* = 0.91; *I^2^* = 78%). It showed that algae intervention significantly decreased HDL-C (WMD = −0.22 mmol/L; 95% CI: −0.38 to −0.06; *p* = 0.008; *I^2^* = 88%) and TC (WMD = −0.45 mmol/L; 95% CI: −0.67 to −0.23; *p* < 0.001; *I^2^* = 88%). This indicated that HDL-C decreased by an average of 0.22 mmol/L in the experimental group after intervention compared with the control group, and that TC decreased by an average of 0.45 mmol/L. The high levels of statistical heterogeneity were found in most of the analysis.

### 3.3. Effect of Algae and Its Extract Intervention on Fast Plasma Glucose and 2-Hour Postprandial Blood Glucose

The effect of algae supplementation on FPG was evaluated in 17 RCTs with 644 subjects. It showed that algae supplementation did not significantly change FPG (Figure 3A: WMD = −0.06 mmol/L; 95% CI: −0.22 to 0.09; *p* = 0.42; *I^2^* = 55%). The results in six RCTs with 201 subjects showed that algae intervention did not significantly change the 2hPBG (Figure 3B: WMD = −0.45mmol/L; 95% CI: −1.85 to 0.96; *p* = 0.53, *I^2^* = 81%).

### 3.4. Effect of Algae and Its Extracts Intervention on HOMA-IR, Insulin and HbA1c

The meta-analysis of nine RCTs with 393 participants (Figure 4A) showed that algae supplementation significantly increased serum insulin (WMD = 5.48 pmol/mL; 95% CI: 3.45 to 7.50; *p* < 0.001, *I^2^* = 78%) and decreased HbA1c (Figure 5A: WMD = −0.18%; 95% CI: −0.27 to −0.10; *p* < 0.001, *I^2^* = 35%). Influence analyses for insulin demonstrated that the heterogeneity belonged to the study of Merhrangiz et al. [42]. Therefore, reanalysis was performed after excluding this study. The inter-study heterogeneity was clearly reduced for insulin (*I^2^* = 51%, Cochrane Q test *p* = 0.05) and it showed that the intervention significantly increased insulin (Figure 4B: WMD = 6.05 pmol/mL; 95% CI: 4.01 to 8.09; *p* < 0.001; *I^2^* = 51%) level. It showed that insulin increased by an average of 5.48 pmol/mL in the experimental group after intervention compared with the control group, and that HbA1c increased by an average of 0.18%. The five RCTs with 236 subjects showed that there was not a significant change in HOMA-IR (Figure 5B: WMD = −0.28; 95% CI: −0.60 to 0.03; *p* = 0.08, *I^2^* = 0%).

### 3.5. Subgroup Analysis

As is shown in Table 2, the algae and its extract intervention significantly increased the levels of insulin in more than 40 subjects and significantly reduced the levels of 2hPBG in Asian participants (Figure 6). The results of subgroup analysis revealed that algae supplement significantly reduced the HDL-C and TC levels in the subgroup with more than 40 persons and the intervention duration more than 10 weeks. It was also found that intervention duration (<10 or >10 weeks) and sample size (<40 or ≥40 people) did not significantly change participants’ FPG, LDL-C and TG levels. Algae intervention significantly reduced HDL-C and TC levels in East Asian people, the spirulina intervention group and both in healthy and obese subjects. The intervention significantly increased TG levels in healthy people but the levels of LDL-C were reversed, and it significantly increased other unhealthy people’s LDL-C levels (Table 2).

### 3.6. Publication Bias

No significant publication bias was found in the inspection of the funnel plot (Appendix A). Similarly, Begg’s ranking correlation and Egger’s linear regression test were also performed to confirm publication bias. Table 3 lists the results of the Begg’s test and the Egger’s test. These results did not show any evidence of publication bias in this analysis (all *p* > 0.05).

## 4. Discussion

Our research showed that algae intervention improved glucolipid metabolism, showing that algae intervention could be effective in improving T2D and CVD. The results of our meta-analysis revealed a significant effect of supplementation with algae and its extracts in reducing HbA1c, TC and HDL-C levels. It also showed significant effects in increasing insulin levels. The combined results are robust and remain significant in the missing sensitivity analysis (Appendix A). Our results are similar to the study performed by Haohai et al. [56]. In addition, another meta-analysis showed similar results [57], but the results of HDL-C were the opposite. Non-insulin-dependent or type 2 diabetes is often a result of prolonged obesity, usually presenting concomitantly with impaired glucose tolerance, hypertension, and hyperlipidemia [10]. Specific goals of medical nutrition therapy for diabetics included reaching and maintaining near-normal blood glucose levels, achieving optimal blood lipid levels, consuming enough calories to achieve a reasonable body weight, and improving overall healthy nutrients by maintaining a balanced macro- and micro-intake [58,59]. Therefore, the algae intervention was beneficial to diabetics by improving their glucose metabolism and lipid distribution indicators. Meanwhile, dyslipidemia, hypertension, hyperglycemia, and high level of HbA1c are considered to be the key risk factors for atherosclerotic cardiovascular disease in humans. There is sufficient evidence to show that the potential role of lipid-lowering treatment interventions is to reduce the risk of atherosclerosis. Seaweed is rich in dietary fiber, and there is evidence that some soluble fibers bind to bile acids or cholesterol during the formation of the micellar cavity [60,61]. The resulting reduction in hepatocellular cholesterol leads to an upregulation of LDL receptors, which increases LDL cholesterol clearance. Edible seaweed is rich in non-starch polysaccharides (dietary fiber), protein, minerals and vitamins. The majority of studies have shown satisfactory results in the use of Undaria spp. in animal studies of diabetes, including improved blood lipids status, reduced inflammatory responses, reduced weight gain, and adjusted blood glucose [62,63]. Other possible mechanisms include inhibition of fermentation products (production of short-chain fatty acids such as acetic acid, butyric acid, and propionate) to synthesize liver fatty acids; changes in intestinal motility; high-viscosity fibers lead to absorption of macronutrients reducing and insulin sensitivity increasing; increased satiety leading to lower energy intake [61]. 

In our subgroup analysis, lipid modification of algae intervention was also found in subjects with > 10 weeks intervention and participants ≥ 40. The subgroup analysis results showed that algae intervention significantly reduced TC and LDL-C levels in healthy people and significantly reduced LDL-C levels in obese and overweight people. This further confirmed the improvement effect of algae intervention on lipid metabolism. The results of HDL-C and its subgroup analysis were contrary to expectations. The previous study [56] summarized the possible links between spirulina intervention and blood lipids. After the inclusion of chlorella and other algae or its extracts, the effect of algae interventions on HDL-C is unclear, so further clinical trials are needed to verify its effect on HDL-C. Similarly, the elevated role of LDL-C in other unhealthy populations and TG in healthy populations was contrary to expectations. Existing studies have included populations with different health conditions, so it is difficult to judge the health effects of algae interventions with different health conditions. We performed a subgroup analysis of people with different health conditions, but the results were not uniform, except for lipid metabolism. This indicates that further clinical studies on different healthy people are needed to explore the effect of algae intervention on lipid metabolism. Meanwhile, it showed significant change in reducing LDL-C levels of the longer intervention time (>10 weeks) and the larger number participants (≥40), which has guiding significance for future clinical trials. For the intervention species, spirulina has the significant effect on reducing TC similar with previous studies [56,57]. Studies have shown that the hypocholesterolemic effects of spirulina concentrates24 may include inhibiting jejunal cholesterol absorption and ileal bile acid reabsorption, as well as increasing fecal cholesterol and bile acid excretion [64,65]. In addition, C-phycocyanin, the main component of spirulina, can reduce lipid concentration through eliminating free radicals, inhibiting lipid peroxidation, inhibiting nicotinamide adenine dinucleotide phosphate oxidase expression, increasing lipoprotein lipase and liver triglyceride lipase, glycated serum protein peroxidase and superoxide dismutase activities [66,67,68]. Moreover, the algae supplement can significantly reduce TC levels in East Asian populations, which suggests that algae intervention may be more effective for East Asians.

The results of our meta-analysis demonstrate that algae and its extract intervention could statistically reduce 2hPBG after treatment in Asian people. It suggested that algae intervention might be more effective for East Asians. For every 1% increase in glycated hemoglobin (HbA1c) concentration, the risk of coronary heart disease increased by 11% [69]. It was reported that consuming diets rich in soluble and insoluble fiber produces satiety, could improve glycemic control and reduce total energy intake, adiposity and blood lipids [70,71]. Edible seaweed is rich in non-starch polysaccharides, protein, minerals and vitamins, while low in lipids, which provides fewer calories [72,73]. Even though seaweed can interfere with the bioavailability of other dietary ingredients [74,75], seaweed polysaccharides, which cannot be fully digested by intestinal enzymes, could be considered to be the sources of dietary fiber. It was hypothesized that the hypoglycemic mechanism of seaweed occurs because the fiber in seaweed delays the absorption of glucose and lipid, thus improving glucolipid metabolism. Meanwhile, spirulina is a rich source of protein and could provide quality protein. Protein and amino acids are known to stimulate insulin production. This effect might be responsible for lowering postprandial blood glucose levels [76,77].

The present meta-analysis has some limitations. Firstly, there is a high degree of heterogeneity between studies, which have not been addressed through extensive subgroup and sensitivity analysis. The source of heterogeneity might be due to differences in study design, the number of participants and baseline characteristics (age, sex, body mass index). Secondly, the quality of the included studies is uneven. Some trials were lacking the information in random sequence generation and types of blinding. These factors may cause imprecise results. Thirdly, although extensive searches and clear inclusion criteria have been developed, we may not have fully identified all relevant articles related to the use of seaweed intervention, especially unpublished trials and grey literature. According to the clinical practice guidelines, LDL-C is considered the main target of lipid-lowering treatment. The lipid-modifying effects of algae were established in our present meta-analysis in TC. However, there was no significant statistical significance in other lipid profiles. Our race was limited to Asian populations, with only three non-Asia studies and there was no population representative. Finally, the biological mechanisms driving metabolic changes may be distinctly different among different studies. At the same time, the degree of processing of the intervention supplements was various in different studies, and the metabolic mechanism was also different. This is the main limitation and one of the sources of our heterogeneity.

To our knowledge, the current meta-analysis is the first to assess the effects of algae and its extracts on glucose and lipid metabolism from RCTs. Grade analyses demonstrated that the quality of the result for all estimates was moderate. The advantage of this meta-analysis is that the results may be reliable because of evidence of low heterogeneity in eligible studies. Therefore, this study can reduce the controversy of the relationship between algae and its extracts and glucose and lipid metabolism.

## 5. Conclusions

The results indicated that seaweed intervention improved levels of insulin and reduced levels of HbA1c and TC levels, but the changes in other lipid profiles, FPG, HOMA-IR were not statistically significant. Our study also demonstrated that an intervention duration of 10 weeks or higher and participants of 40 or more are more useful. It revealed that seaweed intervention may be more effective for East Asians. Moreover, seaweed supplementation is useful for healthy or obese subjects. The combined results showed a significant clinical improvement in CVD and T2D risk. Seaweed consumption may be considered as an adjunct to the prevention and treatment of cardiovascular disease and type 2 diabetes in humans. Therefore, more precise RCTs on subjects with different health status and different races is recommended to clarify the effect of seaweed, seaweed polysaccharide, seaweed polypeptide, algae polyphenol and its products’ intervention on glycolipid metabolism to estimate the effects of CVD and type 2 diabetes.

## Figures and Tables

**Figure 1 nutrients-12-01712-f001:**
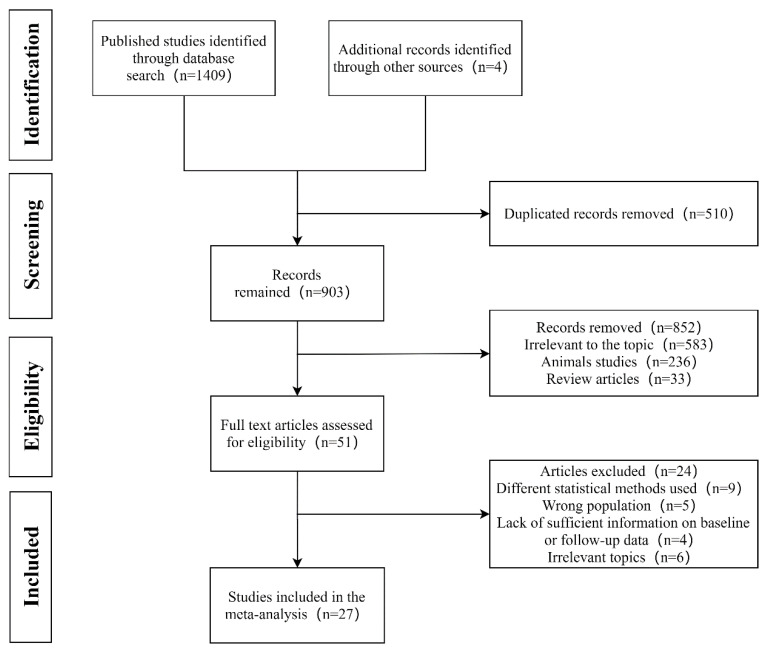
Flowchart of database searches and studies included in the present meta-analysis.

**Figure 2 nutrients-12-01712-f002:**
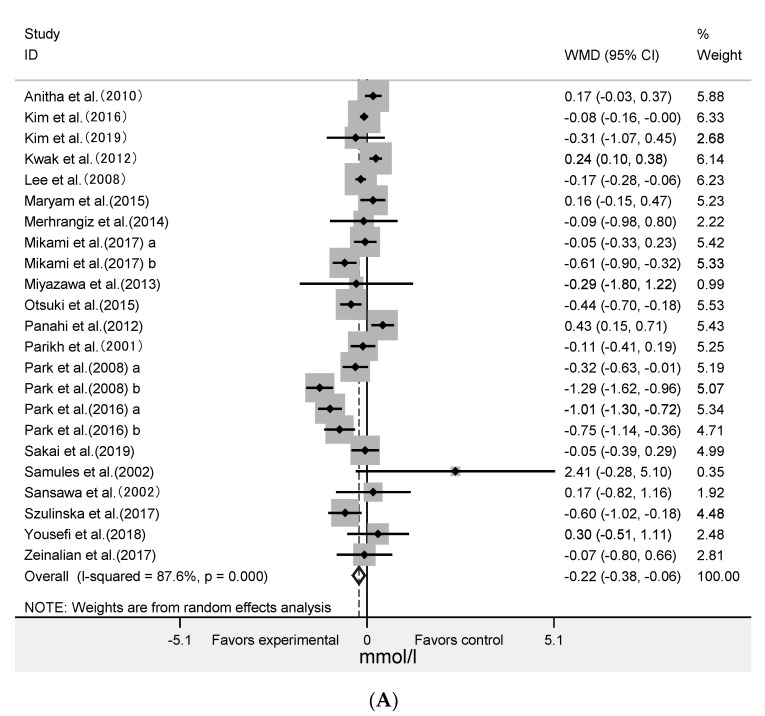
Forest plot of the effect of algae supplementation on high-density lipoprotein cholesterol (**A**), low-density lipoprotein cholesterol (**B**), triglycerides (**C**), and total cholesterol (**D**).

**Figure 3 nutrients-12-01712-f003:**
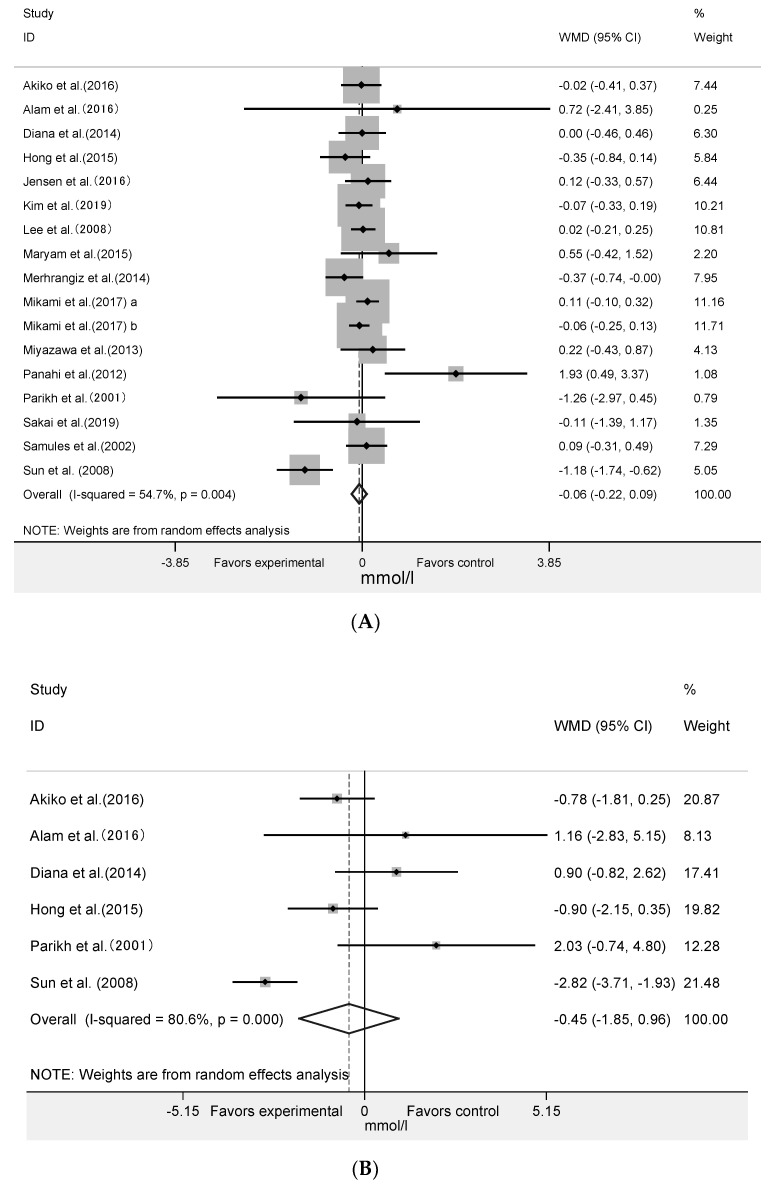
Forest plot of the effect of algae supplementation on fasting plasma glucose (**A**) and 2-h post-meal blood glucose (**B**).

**Figure 4 nutrients-12-01712-f004:**
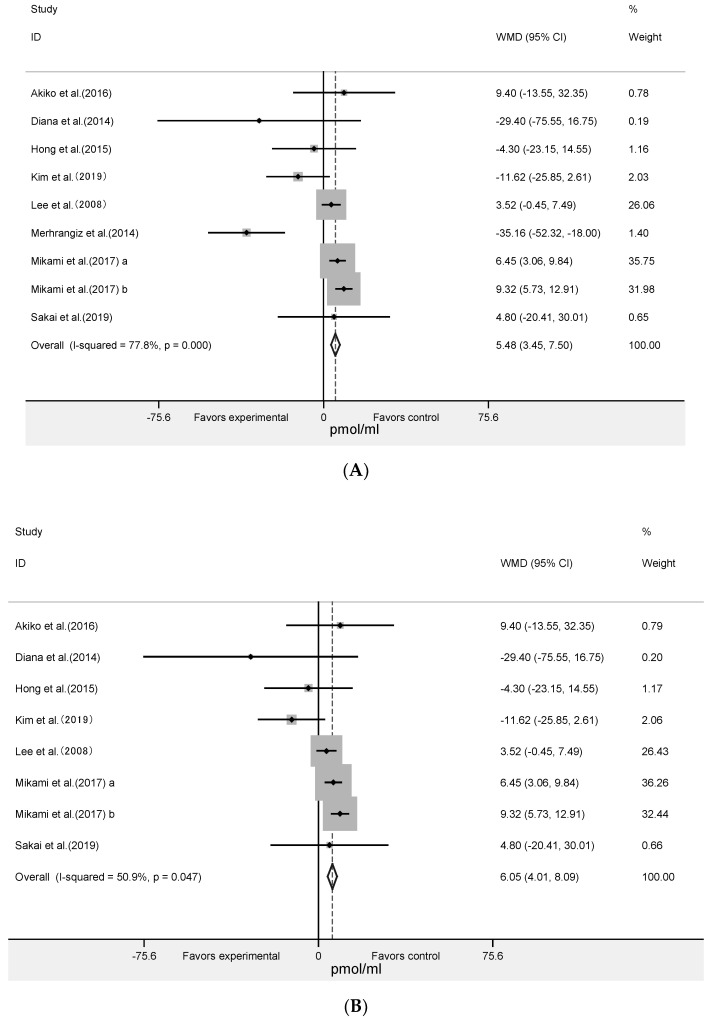
Forest plot of the effect of algae supplementation on insulin (**A**) and Leave-one-out sensitivity analysis of the impact of algae supplementation on insulin (**B**).

**Figure 5 nutrients-12-01712-f005:**
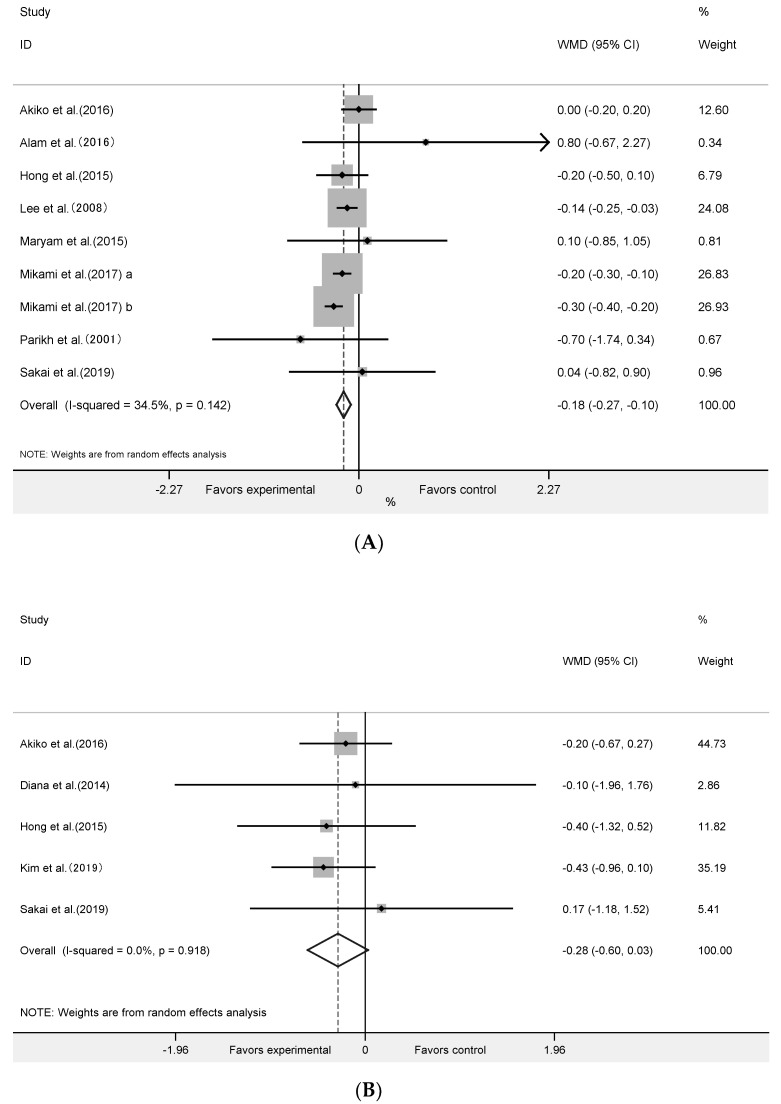
Forest plot of the effect of algae supplementation on glycosylated hemoglobin (**A**) and homeostasis model assessment-insulin resistance index (**B**).

**Figure 6 nutrients-12-01712-f006:**
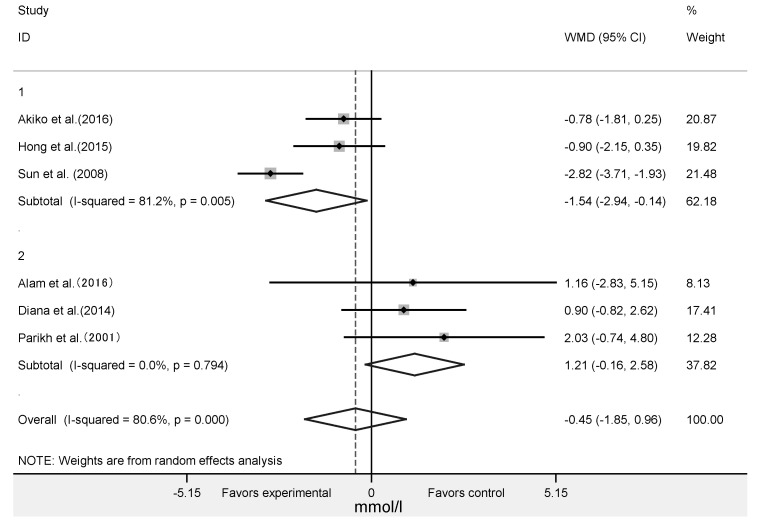
Forest plot of the effect of algae supplementation on 2hPBG in different regions.

**Table 1 nutrients-12-01712-t001:** Characteristics of the included randomized controlled trails (RCTs).

First Author (year)	Country	RCT Design	Sample Number	Male (%)	Age (year)	Health Status	Intervention Duration	Intervention Products	Main Outcomes
Ramamoorthy et al. [51] (1996)	India	NA	20	NA	40–60	Hypercholesterolemia subjects	3 months	Spirulina (2 g/day; 4 g/day)	TG, TC
Parikh et al. [35] (2001)	India	Parallel	25	60%	46–61	patients with type 2 diabetes	2 months	Spirulina tablets (1 g/day)	HbA1c, FPG, 2hPBG, LDL–C, HDL–C, TG, TC
Samules et al. [50] (2002)	India	Parallel	23	74%	3–12	patients with hyperlipidemic nephrotic syndrome	2 months	Spray–dried spirulina capsules (3 g/day)	FPG, LDL–C, HDL–C, TG, TC
Sansawa et al. [37] (2002)	Japan	NA	20	40%	45–64	Hyperlipidemia subjects	3 months	Chlorella (3 g/day)	LDL–C, HDL–C, TG, TC
Lee et al. [34] (2008)	Korea	Parallel	37	54%	49–56	patients with type 2 diabetes	12 weeks	Spirulina pills from freeze–dried spirulina (8 g/day)	HbA1c, FPG, Insulin LDL–C, HDL–C, TG, TC
Park et al. [52] (2008)	Korea	Parallel	43; 36	100%	64–68	males aged 60–87; females aged 60–87	16 weeks	Freezedried spirulina pills (8 g/day)	LDL–C, HDL–C, TG, TC
Sun et al. [31] (2008)	Korea	NA	20	45%	51–58	patients with type 2 diabetes	4 weeks	Pills with sea tangle and sea mustard (48 g/day)	FPG, 2hPBG
Anitha et al. [36] (2010)	India	Parallel	80	100%	45–60	patients with type 2 diabetes	12 weeks	Spirulina capsules and diet modification (1 g/day)	LDL–C, HDL–C, TG, TC
Kwak et al. [46] (2012)	Korea	Parallel	51	39%	30–38	healthy subjects	8 weeks	Chlorella (5 g/day)	LDL–C, HDL–C, TG, TC
Panahi et al. [43] (2012)	Iran	NA	63	27%	51–73	dyslipidemic subjects	8 weeks	Chlorella and atorvastatin (0.6 g/day)	FPG, LDL–C, HDL–C, TG, TC
Miyazawa et al. [45] (2013)	Japan	Parallel	12	58%	50–65	nomal senior subjects	2 months	Chlorella (8 g/day)	FPG, LDL–C, HDL–C, TG, TC
Diana et al. [54] (2014)	Mexico	Parallel	21	29%	38–53	overweight or obese adult	3 months	Fucoidan (0.5 g/day)	FPG, 2hPBG, HOMA–IR, insulin, LDL–C, TG, TC
Merhrangiz et al. [42] (2014)	Iran	Parallel	55	55%	20–50	obese patients with NAFLD	8 weeks	Chlorella (1.2 g/day)	FPG, insulin, LDL–C, HDL–C, TG, TC
Hong et al. [32] (2015)	Korea	Cross–over	73	71%	45–62	pre–diabetic adults	12 weeks	Tablets with AG–dieckol (1.5 g/day)	HbA1c, FPG, 2hPBG, Insulin
Maryam et al. [30] (2015)	Iran	Parallel	49	18%	48–65	patients with type 2 diabetes	12 weeks	Capsules of Aogenizomenon extract (3 g/day)	HbA1c, FPG, LDL–C, HDL–C, TG, TC
Otsuki et al. [44] (2015)	Japan	Parallel	32	41%	45–75	adult subjects	4 weeks	Chlorella (6 g/day)	LDL–C, HDL–C, TG
Akiko et al. [55] (2016)	Japan	Parallel	34	97%	40–56	healthy subjects, BMI ≥ 23	16 weeks	Trehalose (10 g/day)	FPG, 2hPBG, HOMA–IR, insulin, HbA1c
Alam et al. [33] (2016)	India	NA	40	NA	35–54	patients with type 2 diabetes	45 days	Spirulina powder (14 g/day)	HbA1c, FPG, 2hPBG
Jensen et al. [53] (2016)	America	Parallel	24	21%	25–62	adult men and women 25–65 years of age	2 weeks	Phycocyaninenriched aqueous extract from Spirulina platensis (2.3 g/day)	FPG
Kim et al. [47] (2016)	Korea	Parallel	34	12%	22–25	healthy subjects	4 weeks	Chlorella (5 g/day)	LDL–C, HDL–C, TC
Park et al. [39] (2016)	Korea	Parallel	45; 33	NA	64–69	Non–obese subjects; obese subjects	12 weeks	Spirulina (8 g/day)	LDL–C, HDL–C, TG, TC
Mikami et al. [38] (2017)	Japan	Parallel	39; 40	67%; 73%	50–60	obese subjects, BMI ≥ 22	8 weeks	Fucoidan (1 g/day; 2g/day)	FPG, insulin, HbA1c, LDL–C, HDL–C, TC
Szulinska et al. [48] (2017)	Poland	Parallel	50	50%	40–58	subjects with treated hypertension	12 weeks	Spirulina capsules (2 g/day)	LDL–C, HDL–C, TG, TC
Zeinalian et al. [49] (2017)	Iran	Parallel	56	16%	25–43	obese individuals	12 weeks	Spirulina platensis supplement (1 g/day)	LDL–C, HDL–C, TG, TC
Yousefi et al. [41] (2018)	Iran	Parallel	38	18%	31–51	obese and overweight subjects	12 weeks	Spirulina (2 g/day)	LDL–C, HDL–C, TC
Kim et al. [40] (2019)	Korea	Parallel	78	40%	27–46	obese or overweight individuals	12 weeks	Gelidium elegans (1 g/day)	FPG, HOMA–IR, insulin, LDL–C, HDL–C, TG, TC
Sakai et al. [29] (2019)	Japan	Cross–over	30	73%	30–79	patients with type 2 diabetes	28 weeks	Fucoidan (1.62 g/day)	HbA1c, FPG, Insulin, LDL–C, HDL–C, TG, TC

**Table 2 nutrients-12-01712-t002:** Results of the effect of algae and its extracts supplementation on outcomes based on subgroup analyses.

Outcome	Variable	No. of Trials	Effect Size (95% CI)	*p*-Value	*I^2^* (%)
**FPG** **(mmol/L** **)**	Intervention duration				
	<10 weeks	9	−0.087 (−0.362, 0.189)	0.538	71.9
	>10 weeks	6	−0.034 (−0.173, 0.106)	0.636	0.0
	Sample size				
	<40	9	−0.064 (−0.279, 0.151)	0.559	57.8
	≥40	6	−0.073 (−0.327, 0.181)	0.575	53.4
	Intervention species				
	Spirulina	5	0.038 (−0.142, 0.218)	0.680	0.0
	Chlorella	3	0.343 (−0.592, 1.279)	0.472	81.1
	Others	9	−0.121 (−0.327, 0.085)	0.249	61.8
	Health condition				
	Health	3	0.071 (−1.099, 0.342)	0.604	0.0
	Type 2 diabetes	7	−0.313 (−0.817, 0.191)	0.223	69.7
	Obesity	5	−0.047 (−0.181, 0.087)	0.491	22.7
	Other unhealth conditions	2	0.874 (−0.909, 2.657)	0.337	82.7
	Area				
	East Asia	8	−0.011 (−0.111, 0.090)	0.833	0.0
	Non−Asia	2	0.061 (−0.263, 0.385)	0.711	0.0
	Southwest Asia	7	−0.098 (−0.710, 0.515)	0.754	77.7
**Insulin** **(pmol/mL** **)**	Intervention duration				
	<10 weeks	3	−0.636 (−11.138, 9.866)	0.905	92.0
	>10 weeks	6	−0.299 (−7.456, 6.858)	0.935	25.7
	Sample size				
	<40	4	−9.597 (29.892, 10.697)	0.354	90.8
	≥40	5	5.157 (2.611, 7.703)	0.000 *	0.0
**HDL−C** **(mmol/L** **)**	Intervention duration				
	<10 weeks	10	−0.068 (−0.276, 0.139)	0.519	83.8
	>10 weeks	13	−0.329 (−0.595, −0.064)	0.015 *	88.4
	Sample size				
	<40	11	−0.158 (−0.441, 0.124)	0.272	89.5
	≥40	12	−0.298 (−0.507, −0.089)	0.005 *	84.9
	Intervention species				
	Spirulina	11	−0.382 (−0.683, −0.080)	0.013 *	89.9
	Chlorella	7	0.028 (−0.218, 0.274)	0.821	83.6
	Others	5	−0.158 (−0.455, 0.140)	0.299	73.2
	Health condition				
	Health	7	−0.456 (−0.817, −0.094)	0.013 *	94.9
	Type 2 diabetes	5	−0.010 (−0.181, 0.162)	0.912	63.8
	Obesity	7	−0.303 (−0.600, −0.006)	0.046 *	60.0
	Other unhealth conditions	4	0.158 (−0.626, 0.942)	0.692	84.0
	Area				
	East Asia	14	−0.368 (−0.573, −0.163)	0.000 *	90.7
	Non−Asia	1	−	−	−
	Southwest Asia	8	0.164 (−0.009, 0.336)	0.063	31.3
**LDL−C** **(mmol/L** **)**	Intervention duration				
	<10 weeks	10	−0.059 (−0.129, 0.011)	0.101	54.9
	>10 weeks	14	0.129 (−0.276, 0.533)	0.533	98.3
	Sample size				
	<40	11	0.121 (−0.314, 0.557)	0.585	98.8
	≥40	13	−0.028 (−0.094, 0.039)	0.418	42.7
	Intervention species				
	Spirulina	12	0.125 (−0.315, 0.564)	0.579	98.5
	Chlorella	7	−0.043 (−0.129, 0.042)	0.321	46.8
	Others	5	−0.084 (−0.179, 0.011)	0.082	26.3
	Health condition				
	Health	7	−0.132 (−0.224, −0.039)	0.005 *	76.6
	Type 2 diabetes	5	0.036 (−0.670, 0.743)	0.919	98.9
	Obesity	8	0.158 (−0.401, 0.716)	0.581	98.9
	Other unhealth conditions	4	0.119 (0.001, 0.238)	0.048 *	0.0
	Area				
	East Asia	14	0.095 (−0.184, 0.373)	0.701	98.5
	Non−Asia	2	−0.055 (−0.636, 0.526)	0.853	79.5
	Southwest Asia	8	0.042 (−0.059, 0.144)	0.413	0.0
**TG** **(mmol/L** **)**	Intervention duration				
	<10 weeks	7	0.186 (−0.345, 0.717)	0.493	57.6
	>10 weeks	15	−0.112 (−0.551, 0.326)	0.616	82.3
	Sample size				
	<40	11	−0.110 (−1.123, 0.902)	0.831	86.9
	≥40	11	0.116 (−0.058, 0.289)	0.190	22.5
	Intervention species				
	Spirulina	11	−0.266 (−0.940, 0.408)	0.439	87.6
	Chlorella	7	0.151 (−0.155, 0.456)	0.335	46.4
	Others	4	0.030 (−0.549, 0.609)	0.919	0.0
	Health condition				
	Health	8	0.463 (0.017, 0.909)	0.042 *	73.1
	Type 2 diabetes	5	−0.472 (−1.451, 0.507)	0.345	87.9
	Obesity	3	0.091 (−0.782, 0.965)	0.838	0.0
	Other unhealth conditions	6	−0.558 (−1.412, 0.297)	0.201	74.7
	Area				
	East Asia	11	0.183 (−0.366, 0.751)	0.514	86.0
	Non−Asia	2	−0.180 (−0.679, 0.318)	0.478	0.0
	Southwest Asia	9	−0.294 (−0.848, 0.259)	0.297	62.6
**TC** **(mmol/L** **)**	Intervention duration				
	<10 weeks	9	−0.128 (−0.383, 0.127)	0.325	79.6
	>10 weeks	16	−0.646 (−0.991, −0.300)	0.000 *	88.0
	Sample size				
	<40	11	−0.242 (−0.513, 0.030)	0.082	83.5
	≥40	14	−0.700 (−1.101, −0.299)	0.001 *	90.0
	Intervention species				
	Spirulina	13	−0.803 (−1.223, −0.383)	0.000 *	90.1
	Chlorella	6	−0.016 (−0.310, 0.278)	0.914	79.0
	Others	6	−0.242 (−0.518, 0.034)	0.085	48.6
	Health condition				
	Health	5	−0.692 (−1.269, −0.115)	0.019 *	94.3
	Type 2 diabetes	5	−0.072 (−0.250, 0.107)	0.432	0.0
	Obesity	9	−0.346 (−0.553, −0.140)	0.001 *	59.2
	Other unhealth conditions	6	−0.810 (−2.137, 0.516)	0.231	94.0
	Area				
	East Asia	13	−0.475 (−0.723, −0.226)	0.000 *	88.0
	Non−Asia	2	−0.200 (−0.507, 0.107)	0.202	0.0
	Southwest Asia	10	−0.582 (−1.276, 0.113)	0.101	89.6

Notes: * Indicates a significant result.

**Table 3 nutrients-12-01712-t003:** Publication bias in the meta-analysis of studies.

Outcomes	Begg’s Rank Correlation Test		Egger’s Linear Regression Test			
	Z Value	*p*-Value	Intercept (95% CI)	t	df	*p*-Value
FPG	0.29	0.773	−0.04 (−1.51, 1.44)	−0.05	16	0.959
2hPBG	1.13	0.260	3.77 (−0.71, 8.24)	2.34	5	0.080
HOMA−IR	0.24	0.806	0.40 (−1.26, 2.06)	0.76	4	0.503
Insulin	0.73	0.466	−1.92 (−4.02, 0.19)	−2.16	8	0.068
HbA1c	0.89	0.371	1.34 (−0.86, 3.54)	1.41	9	0.197
HDL−C	0.21	0.833	−0.88 (−2.84, 1.07)	−0.94	22	0.360
LDL−C	0.72	0.472	0.66 (−3.69, 5.02)	0.31	24	0.756
TG	0.11	0.910	−0.26 (−1.81, 1.28)	−0.36	22	0.726
TC	0.61	0.544	−0.89 (−2.65, 0.87)	−1.04	24	0.307

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
