# Peer review of "Quantifying the Effect of Supplementation with Algae and Its Extracts on Glycolipid Metabolism: A Meta-Analysis of Randomized Controlled Trials"

_nutrients, 2020, doi:10.3390/nu12061712_

Round 1
Reviewer 1 Report
The manuscript describes a systemic literature search for published studies investigating the effects of seaweed or seaweed-derived extracts as dietary supplements to alter glucose and lipid metabolism in humans using randomised control trials. The data were then combined and a meta-analysis was carried out. Some effects were observed that reached statistical significance.
The study is ambitious in its scope, including a wide range of interventions and study designs. The issue of heterogeneity is acknowledged in the Discussion, however the extent to which it limits interpretation is not. The concern I have in interpreting the results is that the biological mechanisms driving metabolic changes are likely to be distinctly different among the different studies. For example, interventions include whole spirulina (a cyanobacterium containing only low levels of indigestible polysaccharide), fucoidan (essentially a pure form of indigestible polysaccharide from brown macroalgae) and whole chlorella (a type of single-celled algae possessing a cell wall impervious to human digestive enzymes). The extent of processing of the dietary supplements are not described or taken into account in the study. These interventions deliver a wide range of different putative active components and most likely involve diverse mechanisms of action. I wonder if the scope of the study is too ambitious.
Publication bias is also addressed in the Discussion, but only from an overall statistical perspective. It seems that studies of spirulina are predominant in the dataset, and moreover, they appear to have created some bias. For example, the significant effect on HDL-C described in section 3.2 appears to be due to four studies from a single research group working on spirulina (Figure 2 A).
When considering all the significant findings as a whole, there is no clear and consistent pattern of metabolic intervention emerging from the data. Therefore it is not possible to gain insight from the meta-analysis.
Besides there overall impressions, there are a number of more specific issues.
- The statistical methods used to combine the data from the different studies is not described. The text merely states "Otherwise the fixed effects modelling method was applied". This is insufficient to understand how the data were combined, what statistical package was used, what assumptions were made and what parameters were used.
- The text is overly general in the use of "seaweed" to describe the interventions. Spirulina and chlorella are not seaweeds. Seaweeds are macroalgae. Some of the interventions are not seaweeds but are extracts from seaweed.
- The figures are not sufficiently explained. These include the meanings of the different symbols, shading and the units of the horizontal axes.
- WMD values are given, along with CI and P values, but not the relative change. It would be useful to clearly describe how substantial these changes are and thereby allow the reader to judge what effect they might have on metabolism.
- The assumptions and procedure for the sub-group analysis should be better explained. The figure is ambiguous as to what each section represents.
- The text contains a regular sprinkling of minor grammatical errors. I did not systematically check the references but I noted that reference 57 is referred to in the text as Huang and in the reference list as Haohai. There are also instances of confusing description, for example in the last two sentences of sectin 3.5, what does "reversed" mean in this context? Who are "other unhealthy" people?
Reviewer 2 Report
This is an excellent article, very interesting, well written and well preform of the other wise sacttered information about the health benefits of seaweed and its extracts. This work help clear what is somewhat scientifically validated claims among a large amount of non scientifically claims done by some manufacturers. It also lays the foundations for future studies using the gold standard, double blind placebo controlled and randomized trials, before firm unequivocual conclusions about the situability of seaweed or its extracts to improve health conditions. Besides, a high number of variables should be taken into account.
Reviewer 3 Report
I have now thoroughly read your manuscript entitled "Quantifying the effect of supplementation with seaweed and its extracts on glycolipid metabolism: a meta-analysis of randomized controlled trials". Mainly, I think that this is a well-performed analysis with some interesting findings.
However, I have one major concern, namely that Spirulina and Chlorella are not classified as seaweeds, but cyanobacteria/microalgae. And, since most of the studies included in the article involves these two species, the title of the paper is misleading and should be changed.
Otherwise, I have no major objections. It would probably be wise to have a native English speaking person read the paper and make sure that minor errors are corrected.
